# Development of a minimal KASP marker panel for distinguishing genotypes in apple collections

Mark Winfield[1]*, Amanda Burridge[1], Matthew Ordidge[2], Helen Harper[1], Paul Wilkinson[3], Danny Thorogood[4], Liz Copas[5], Keith Edwards[1], Gary Barker[1]

1 School of Biological Sciences, University of Bristol, Bristol, United Kingdom, 2 School of Agriculture, Policy and Development, University of Reading, Reading, United Kingdom, 3 Department of Functional and Comparative Genomics, Faculty of Health and Life Sciences, University of Liverpool, Liverpool, United Kingdom, 4 Institute of Biological, Environmental and Rural Sciences, Aberystwyth University, Aberystwyth, United Kingdom, 5 Lullingstone, Fore Street, Winsham, Somerset, United Kingdom

* mark.winfield@bristol.ac.uk

**Data Availability Statement:** All relevant data are within the paper and its Supporting information files.

## Abstract

Accurate identification of named accessions in germplasm collections is extremely important, especially for vegetatively propagated crops which are expensive to maintain. Thus, an inexpensive, reliable, and rapid genotyping method is essential because it avoids the need for laborious and time-consuming morphological comparisons. Single Nucleotide Polymorphism (SNP) marker panels containing large numbers of SNPs have been developed for many crop species, but such panels are much too large for basic cultivar identification. Here, we have identified a minimum set of SNP markers sufficient to distinguish apple cultivars held in the English and Welsh national collections providing a cheaper and automatable alternative to the markers currently used by the community. We show that SNP genotyping with a small set of well selected markers is equally efficient as microsatellites for the identification of apple cultivars and has the added advantage of automation and reduced cost when screening large numbers of samples.

## Introduction

Accurate identification of plant material within germplasm collections is of utmost importance, especially for vegetatively propagated crops which are expensive to maintain. Genotyping of plant genetic resources, therefore, is an essential task for any gene bank to ensure the preservation of maximum genetic diversity whilst, at the same time, avoiding the waste of resources by carrying unknown duplicates. Without careful curation, over time accessions may lose their identity or become mislabelled [1–3]. In addition to phenotyping to determine trueness-to-type, cultivars need to be genetically fingerprinted using an appropriate tool. In apple, traditionally this tool has been microsatellite markers [2, 4, 5]. While an advantage of microsatellites is that they are multi-allelic, co-dominant markers, they are difficult to automate and score. As such, in many crop species microsatellites have been replaced with Single Nucleotide Polymorphism (SNP) markers which are more amenable to high throughput

**Funding:** KE, HH Project 98 Bristol Centre of Agricultural Innovation (BCAI) http://www.bristol. ac.uk/biology/bcai/ The funders had no role in study design, data collection and analysis, decision to publish, or preparation of the manuscript.

**Competing interests:** The authors have declared that no competing interests exist.

automation. Large numbers of SNPs, and the high-density arrays based on them, are available for most of the important crop species including wheat [6], barley [7], potatoes [8] and apples [9, 10]. Datasets generated by these assays provide publicly accessible SNP information for hundreds or thousands of varieties across tens of thousands of loci. Such high-density arrays are widely used by academics and breeders to characterise crop varieties across their entire genomes but cost in the order of £50 per sample. In addition, such procedures require significant bioinformatics support, and are thus too expensive to use for basic identification of the accessions held in germplasm collections. A lower cost, automatable assay capable of replacing the existing apple microsatellite markers would be a valuable tool for germplasm management. To achieve this, we developed a bioinformatics pipeline able to identify, amongst a large number of SNPs, a small number of highly informative markers. These were then converted into high throughput Kompetitive Allele Specific Polymorphism (KASP) markers. To test the efficiency of these markers we have used them to characterise leaf samples collected from over 2,500 individual apple trees: the majority of the samples were collected from the National Fruit Collection (NFC) at Brogdale, Kent, and the Welsh Botanical Gardens at Carmarthen. We have used these two, well curated collections and the MUNQ codes [11, 12] (*Malus* UNiQue codes: unique codes assigned to apple varieties with unique genotypic profiles) assigned to the accessions therein to challenge our SNP panel and test its ability to distinguish all unique accessions and reveal all replicates be they explicitly declared or hidden behind synonyms or because of errors in naming. We believe that these KASP markers offer the apple community a cheap and efficient genotyping tool which, while being as useful as the existing microsatellites at identifying varieties, offers the potential for screening very large numbers of varieties at a considerably reduced cost. This approach, we believe, will be equally applicable for low cost varietal identification in other crop species.

## Materials and methods

### Collection of plant material

In response to the request to supply field permits for work carried out at Brogdale and the Welsh Botanic Garden, we can report that these were not required. The curators of these two collections are included amongst the authors of the manuscript.

In total, 2,675 leaf samples were collected from 2,652 individual trees (24 technical replicates were collected form the variety Yeovil Sour) representing more than 2,200 named varieties. Principally, these came from the National Fruit Collection at Brogdale in Kent (2,104 trees sampled) and the Welsh Collection at the National Botanic Garden of Wales in Llanarthney (143 trees samples). Samples were collected from a further 428 trees from various orchards and gardens: these included, a small number of trees from small, local orchards in Bristol, and all the samples (372) used in the study of Harper et al. [13] (S1 File: spreadsheet '*Samples studied*'). Based on name alone, the samples represented 2,287 different varieties: for 2,058 of these varieties apparently a single tree was sampled (these are referred to as 'nominal singletons'); for the other varieties, again based solely on name, we had samples taken from more than one tree (often these were from different orchards or gardens). Sampling took place in June / July (2018) and June / July (2019) so that features of the fruit could be observed.

In all cases, three leaf discs (diameter of 0.7 cm) from a single young leaf were sampled and placed in a 96 well plate. Plates were stored at -70˚C until further processing could take place.

### DNA extraction

DNA extraction was carried out as previously described [14]. Purification by Qiagen column was found to be a necessary step to reduce the levels of phenolic compounds [15] and so obtain

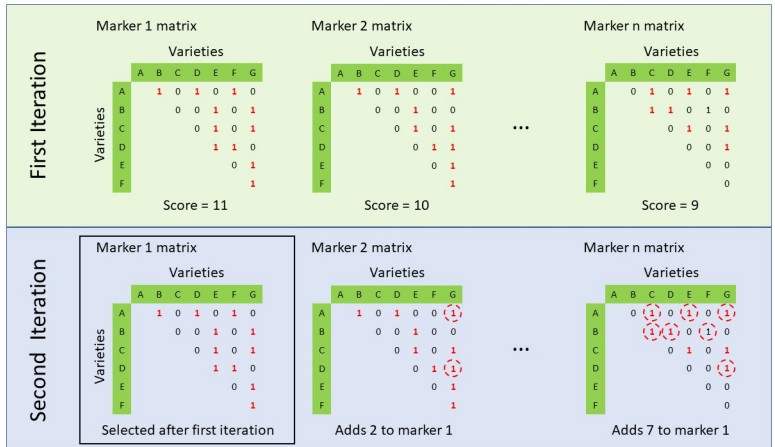

**Fig 1. A bioinformatics approach to identify an optimal, minimal set of SNP markers; here, varieties A—G have been genotyped with *n* SNP markers.** For each marker, a matrix is produced that shows which varieties are discriminated with that SNP: discriminated (1) or the same (0). In the first iteration, of the three markers shown, marker 1 performs best as it discriminates 11 of the variety pairs. Therefore, marker 1 is selected as our first marker and we repeat the process in the second iteration. All other markers are evaluated to see what additional varieties they would discriminate. These are circled in the matrixes for marker 2 and marker *n*. Although marker *n* had the lowest score in the first round, it adds more discrimination than marker 2, so this marker would be selected along with Marker 1 and carried forward to the third iteration. This process is repeated until every variety is resolved (matrix has no zeros remaining) or adding more markers does not improve discrimination further.

accurate genotyping results. DNA was diluted with PCR-grade water to an average of 2.5 ng per reaction.

## Identification of the SNP marker set

We used a Perl script (available at https://github.com/pr0kary0te/minimalmarkers.git) together with test data from Harper *et al.* [13], to identify a set of 21, highly polymorphic SNP markers capable of differentiating all 372 cultivars used in that study. The script first selects the marker with the highest minor allele frequency then evaluates all remaining markers to see which one differentiates the highest number of cultivars that were not split by the first marker. The script iterates this process until a point where adding more SNPs provides no further splits or where all cultivars are resolved (Fig 1). Additional markers were added manually from the list sorted by differentiation score by the above script with the aim of maximising the number of chromosomes covered. Initially, each chromosome had at least one marker.

## Genotyping protocol

For each of the 21 markers identified by the perl script (see above), two allele-specific forward primers and one common reverse primer were designed using the same primer sequence used in Harper *et al.* [13]; (primer sequences are available in S1 File: spreadsheet '*KASP probes*'). Genotyping was performed using the KASP™ system (LGC Genomics) scaled for 1,536 format. Each reaction was performed using 2.5 ng DNA, 0.5µL KASP reaction mix, 0.018 µL assay mix (12 µM of each forward primer, 30 µM reverse primer) in a total volume of 1 µL. Amplification was performed using a Hydrocycler-16 (LGC, Genomics) under the following conditions: 94°C for 15 minutes; 10 cycles of 94°C for 20s, 65–57°C for 60s (dropping 0.8 C per cycle); 35 cycles 94°C for 20s, 57°C for 60s. Fluorescence detection was performed using a BMG

Pherastar® scanner and genotype calling performed using the Kraken software package (LGC Genomics).

## Concordance with SeqSNP markers

The KASP markers used in this study had been converted from markers based on the SeqSNP® protocol and 372 samples analysed here had been genotyped previously using that platform. A simple python script was used to determine concordance between results from the two platforms.

## Estimating error rate and identification of mislabelled samples

Replicate samples were expected to have 21 identical SNP calls; any SNP difference between 'replicates' was assumed, therefore, to be the result of an error in calling, although which sample bore the error could not be determined (Fig 2). Thus, for a variety for which we had two samples (assuming that these were true biological replicates), one marker difference in SNP profile would represent an error rate of 2.38% (1 / 42). For varieties with three or more samples the consensus call was assumed to be correct and the others compared with it; thus, for three samples, there are 3 x 21 comparisons and a single SNP difference in one of them would represent a 1.58% (1 / 62) error rate. The total error rate was determined by dividing the total number of errors by the total number of SNP calls. Two estimates of error were made: 1) an estimate under the assumption that all nominal replicates were, indeed, genuine biological replicates and so all errors were counted, regardless of the number of SNP differences between them; 2) an estimate under the assumption that only samples that had fewer than 3 SNP differences could be taken as genuine biological replicates. Most of the accessions obtained from Brogdale and the Welsh Collection have been genotyped previously using microsatellite or DArT markers or both and had been assigned MUNQ codes; additional MUNQ codes, created as part of an extension of the worked initiated in [11, 16], were provided by Caroline Denancé, INRAE, Angers, France [12]. These codes were taken into account where possible when estimating error rate in SNP calling.

| Accession | BA001 | BA003b | BA004 | BA005 | BA006 | BA007b | BA008 | BA012 | BA014 | BA015 | BA017 | BA018b | BA019b | BA021 | BA022 | BA023 | BA024b | BA026 | BA027b | BA029 | BA031 |
|---|---|---|---|---|---|---|---|---|---|---|---|---|---|---|---|---|---|---|---|---|---|
| Falstaff 1 | 2 | 1 | 1 | 1 | 2 | 0 | 1 | 2 | 1 | 2 | 0 | 1 | 2 | 1 | 0 | 0 | 2 | 2 | 0 | 0 | 1 |
| Falstaff 2 | 2 | 1 | 1 | 1 | 2 | 0 | 1 | 2 | 1 | 2 | 0 | 1 | 2 | 1 | 0 | 0 | 2 | 2 | 0 | 0 | 1 |
| Kim 1 | 1 | 2 | 1 | 1 | 1 | 1 | 2 | 2 | 0 | 1 | 1 | 0 | 2 | 2 | 0 | 1 | 2 | 0 | 1 | 2 | 0 |
| Kim 2 | 1 | 0 | 1 | 2 | 0 | 2 | 0 | 2 | 2 | 1 | 0 | 0 | 1 | 2 | 1 | 0 | 2 | 0 | 2 | 2 | 0 |
| Ashton Bitter 1 | 0 | 1 | 0 | 2 | 1 | 1 | 1 | 2 | 1 | 2 | 0 | 1 | 1 | 1 | 2 | 1 | 1 | 0 | 2 | 2 | 9 |
| Ashton Bitter 2 | 0 | 1 | 0 | 2 | 1 | 1 | 1 | 2 | 1 | 2 | 0 | 1 | 1 | 1 | 2 | 1 | 1 | 0 | 2 | 2 | 1 |
| Ashton Bitter 3 | 0 | 1 | 0 | 2 | 1 | 1 | 1 | 2 | 1 | 2 | 1 | 1 | 1 | 1 | 2 | 1 | 1 | 0 | 2 | 2 | 1 |
| Gavin 1 | 0 | 1 | 1 | 1 | 2 | 1 | 0 | 1 | 1 | 1 | 0 | 0 | 2 | 1 | 1 | 1 | 2 | 0 | 1 | 1 | 2 |
| Gavin 2 | 0 | 1 | 1 | 1 | 2 | 1 | 0 | 1 | 1 | 1 | 0 | 0 | 2 | 1 | 1 | 1 | 2 | 0 | 1 | 1 | 2 |
| Gavin 3 | 2 | 2 | 1 | 1 | 1 | 2 | 1 | 1 | 0 | 2 | 0 | 0 | 1 | 1 | 1 | 0 | 2 | 1 | 1 | 1 | 1 |

**Fig 2. 'Error' calling for SNP markers.** SNP profiles for the two samples of Falstaff are identical (0% error) whilst those for the two samples of Kim differ by 12 calls (12 / 42 = 28.6% error). One of the three samples of Ashton Bitter differs from the other two by one call, but one of the samples has a missing call (blue) so that only 62 calls were made (1 / 62 = 1.6% error). One of the three samples of Gavin has 9 differences from the other two (9 / 63 = 14.3% error). The total 'error' for these 10 lines, therefore, is 22 / 209 = 10.5%. However, regardless of the names given to accessions, if one considers samples that differ by more than two SNP calls to be different varieties, then there is only 1 error, that for Ashton Bitter 3; the two samples of Kim are not biological replicates and we only have two samples of Gavin. The error rate for the remaining 7 samples, then, is (1 / 146 = 0.6% error).

## Dimensionality reduction

The relationship between the accessions was determined from the SNP data. A pair-wise similarity matrix was constructed using a custom Python script (available on request); similarity was calculated as the number of calls in common between two accessions divided by total number of markers scored for them; markers that had missing calls for either of the accessions being compared were not used to estimate similarity. The resulting matrix was imported into the R statistical software package version 3.3.1 [17]; multi-dimensional scaling was performed using 'mdscale'; dendrograms were created using the '*hclust*' function and converted to phylograms using the '*as.pyhlo*' function of the ape library.

The resulting dendrogram was bootstrapped using the library pvclust [18].

The final, bootstrapped dendrogram was exported as a nexus file and manipulated in the web-based program iTOL (Interactive Tree of Life) [19]; interactive dendrograms of the data set are available at http://itol.embl.de/shared/MarkW58.

## Calculation of heterozygosity levels

Levels of heterozygosity were calculated for all samples based on 21 markers by dividing the number of heterozygous loci by the total number of genotyped loci.

## Statistical analysis

One-way ANOVA to determine the relationship between recorded ploidy and heterozygosity was performed in the R statistical software package version 3.3.1 [17] using the library FSA and the Tukey test using the HSD test from the library agricolae.

## Results

### Developing a minimum set of SNP markers

Following an earlier study of 372 accessions with 1,300 SNP markers [13], we considered the possibility of identifying a minimum set of markers that would be sufficient to identify all cultivars studied. To achieve this, we used a bioinformatics approach (Fig 1). Initially, we identified a set of 31 markers for which we designed KASP forward and reverse primers. Following preliminary experiments, these 31 markers were reduced to 21 KASP probe assays which showed a high level of reproducibility and good separation between homozygote and heterozygote genotypes. The sequences of these 21 KASP probe, along with the chromosomes to which they are assigned, are given in S1 File: spreadsheet '*KASP probes*'.

### Genotyping accuracy

In total, the DNA from 2,675 leaf samples (2,651 individual accessions plus 24 technical replicates from the variety Yeovil Sour) were genotyped with 21 markers. The average fail rate (marker produced no signal or was not called (Fig 3)) across all 21 markers was 3.5%. However, markers had quite different fail rates (Table 1): the best performing marker, BA03b, had a fail rate of only 1.6% (42/2,675) whilst the worst, BA08, had a fail rate of 6.9% (185/2,675). The number of markers that failed per sample was also variable: most samples (1,903, 71.0%), had no failed markers whilst one sample, Salome, had 21 that failed (Table 2). Only those accessions that had calls from at least 19 of the 21 SNP markers were analysed further. As a consequence, 193 samples, including one of the technical replicates of Yeovil Sour, were removed from the study and only 2,482 samples were analysed further. A list of the accessions retained along with their SNP profiles is available in S1 File: spreadsheet '*Samples studied*'. All further analysis is based exclusively on these 2,482 samples.

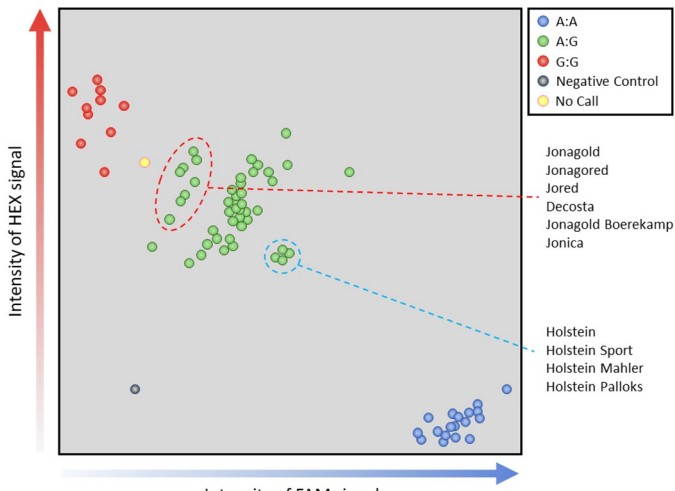

**Fig 3. SNP calling for KASP marker BA08.** Sample position is determined by intensity of signal detected from fluorochromes bound to allele-specific primers, A on the X-axis, G on the Y-axis. Samples are classed by cluster: blue = A:A; green = A:G; red = G:G. To avoid miscalls, samples that fall between clusters (yellow) will not be assigned a call. Intensity ratios within clusters may differ and this may reflect some underlying characteristic of the samples (e.g., ploidy or quality of the DNA) or hybridisation efficiency of primers due variation in sequences surrounding the targeted SNP. The genotypes ringed in red are all mutations of Jonagold, those ringed in blue are mutations of Holstein. (Image redrawn).

## Number of distinct SNP profiles

The names of the 2,482 accessions analysed indicate that there were 2,123 distinct varieties in our study (1,919 singletons and 248 with replicates); (S1 File: spreadsheet '*Samples with Nominal Reps*'). Using a simple python script to count distinct SNP profiles we identified 1,850 SNP profiles: 1,520 profiles were unique to a single sample, whilst 310 were shared by at least two samples (total number of samples with shared profiles was 962) (the number of samples sharing the same SNP profile ranged from 2–31). These figures, however, overestimate the number of singletons and underestimate the number of samples with biological replicates because miscalls create false, 'novel' SNP profiles. One can adjust for these experimental errors when samples have known biological replicates, e.g. the sample Black Vallis 1 differed by one SNP (BA14) from the other two Black Vallis samples creating two distinct genotypes where we presume there should be only one. However, unflagged samples that are identical (e.g. synonyms) but differ because of an incorrect call will go unidentified (we can't know what the missing value would have been so can't unite the pair). Having adjusted, where possible, for these issues, we identified 1,820 SNP profiles: 1,495 singletons, and 330 that are shared by two or more samples representing 987 accessions in total.

## Visualisation of the relationship between the apple varieties genotyped

The purpose of this work was to develop a small set of highly informative markers rather than carry out a phylogenetic study as, based upon the number of markers used, we acknowledge that any given dendrogram would not be a true representation of the relationship between the accessions studied. However, for the sake of description we present a dendrogram (an interactive, bootstrapped dendrogram can be accessed at http://itol.embl.de/shared/MarkW58): bootstrap values (1000 reps) only support clusters of individuals that differ from each other by a

**Table 1. Number of fails per marker.**

| Marker ID | Number of Fails | |
|---|---|---|
| | **Before (2,675)** | **After (2,482)** |
| BA001 | 60 (2.2%) | 24 (1.0%) |
| BA003b | 42 (1.6%) | 15 (0.6%) |
| BA004 | 71 (2.7%) | 21 (0.9%) |
| BA005 | 59 (2.2%) | 14 (0.6%) |
| BA006 | 72 (2.7%) | 26 (1.1%) |
| BA007b | 52 (1.9%) | 17 (0.7%) |
| BA008 | 185 (6.9%) | 101 (4.2%) |
| BA012 | 67 (2.5%) | 22 (0.9%) |
| BA014 | 92 (3.4%) | 28 (1.1%) |
| BA015 | 69 (2.6%) | 29 (1.2%) |
| BA017 | 112 (4.2%) | 42 (1.7%) |
| BA018b | 115 (4.3%) | 58 (2.4%) |
| BA019b | 73 (2.7%) | 25 (1.0%) |
| BA021 | 136 (5.1%) | 48 (2.0%) |
| BA022 | 147 (5.5%) | 69 (2.9%) |
| BA023 | 50 (1.8%) | 17 (0.7%) |
| BA024b | 95 (3.6%) | 39 (1.6%) |
| BA026 | 45 (1.7%) | 12 (0.5%) |
| BA027b | 79 (3.0%) | 41 (1.7%) |
| BA029 | 74 (2.8%) | 37 (1.5%) |
| BA031 | 102 (3.8%) | 49 (2.0%) |

In the whole data set (2,675 samples assayed with 21 markers) 56,175 data points were produced; the total number of failed was 1,797 or 3.2%. After remove of the 193 samples for which more than two markers failed, the corresponding numbers were 735 or 1.4%.

maximum of 1 or 2 SNP markers; apparent clusters based on greater difference than that were not supported (Fig 4, http://itol.embl.de/shared/MarkW58).

## Concordance with SeqSNP calls

Numerous samples assayed in this study (372) had been genotyped previously using the SeqSNP® protocol [13]. Given that the 21 KASP markers were designed directly from the SeqSNP markers used in that study, we expected complete concordance between the SNP profiles produced by the two assays. However, overall concordance between the two platforms for

**Table 2. The number of marker failures per number of apple accessions.**

| No. of Fails | No. of Accessions | Percentage |
|---|---|---|
| 0 | 1,903 | 71.1 |
| 1 | 424 | 15.9 |
| 2 | 155 | 5.8 |
| 3 | 193 | 7.2 |

Only samples for which 19 or more markers gave a score were analysed further.

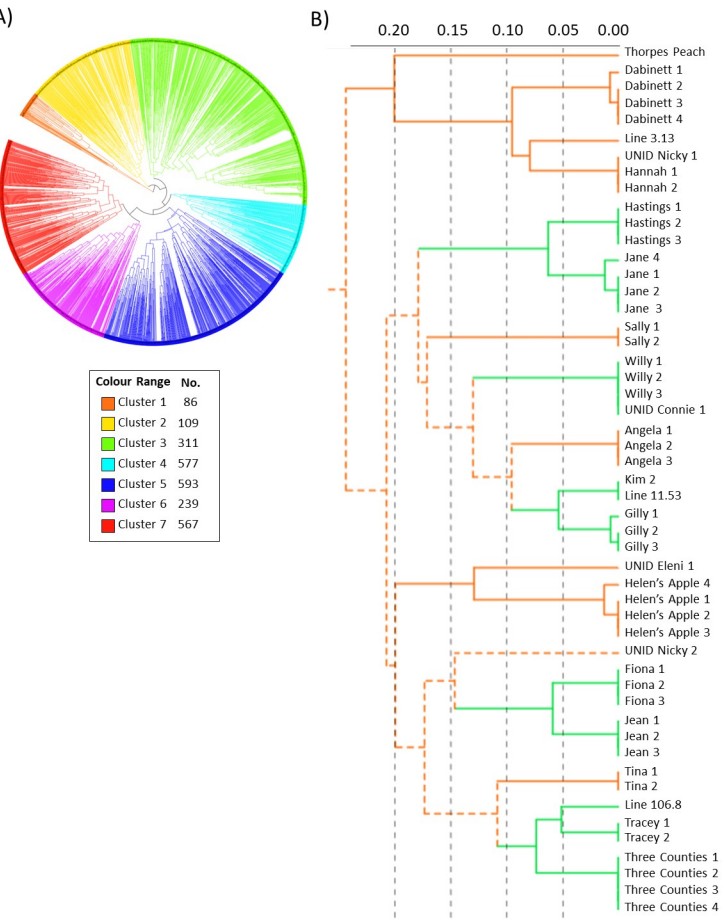

**Fig 4. A)** Dendrogram of 2,482 apple samples divided into seven clusters; the number of samples in each cluster is shown in the legend. **B)** Detail of Cluster 1: solid lines indicate clusters for which the branch had 98% bootstrap support (colour of clusters alternates to assist viewing); black, horizontal dashed lines show distance (a distance of 0.05 corresponds to 1 SNP difference in the SNP profile).

all 21 markers was only 95.8%. For three of the 21 KASP markers (BA06, BA12 and BA31) there was total concordance. For a further twelve markers there were 6 or fewer differences between the calls given by the two platforms (98.4–99.7% concordance); a further four had 17 or fewer differences between the two platforms (95.6–97.4% concordance). Thus, the average concordance between the two platforms for these 19 markers was 98.9%. Two of the markers however, BA07b and BA22, showed much less agreement. The SeqSNP® marker MDC009399-297 reported an A/G SNP. The KASP marker derived from it, BA07b, was designed to report the opposite strand of the DNA so was a C/T SNP; heterozygotes A:G should thus be C:T and the homozygotes A:A and G:G should be T:T and C:C, respectively. All the original homozygote scores (A:A = 90 and G:G = 29) concorded, on this basis, between the two marker systems; however, 71 of the 257 markers that gave a heterozygous call (A:G) with SeqSNP® were called homozygotes (C:C) with the KASP markers. The SeqSNP® marker MDC004696-296, a C/T SNP, gave calls of C:C = 18, C:T = 339 and T:T = 21; the KASP marker, BA22, designed from it gave calls of C:C = 199; C:T = 158 and T:T = 21. That is, 181 of

the genotypes that were called heterozygote by SeqSNP® were now called homozygote by KASP.

An alternative approach to compare results from the SeqSNP® and KASP platforms was to compare the relationships of the 372 lines that have been genotyped on both. Using this approach, accessions that, within the boundaries of errors, were identical when analysed with 1,300 SeqSNP® markers were expected have identical SNP profiles for the 21 KASP markers used here. Essentially, this proved to be the case; all 78 clusters produced using 1,300 SeqSNP® markers were also produced using KASP markers (100% bootstrap support from 1000 iterations). The only differences were two highly supported clusters (Devon Winter Stubbard and Stubbard; Red Jersey 2 and Unid Burrowhill Early 1) produced with the 21 KASP markers that were not supported by SeqSNP® (http://itol.embl.de/shared/MarkW58; *bootstrapped_21_1000.nex and bootstrapped_1276_1000.nex*).

## Accuracy of scoring replicates

The samples with technical or biological replicates allowed us to estimate the error rate of our SNP assays. For example, from the variety Yeovil Sour we collected 24 independent samples (only 23 were carried forward because one sample failed genotyping) which we processed and genotyped separately; all proved to have identical SNP profiles. When accession name was used exclusively to identify potentially replicate accessions, 1,919 were noted to be singletons whilst the remaining 563 had at least one replicate—these comprised 204 groups ranging in number from 2 to 23 (S1 File: spreadsheets '*Samples studied*' and '*Samples with Nominal Reps*'). For these samples, the putative biological replicates were collected from different trees, often from different orchards or gardens. The ten nominal replicates of Cox's Orange Pippin were identical to each other, as were the eight nominal replicates of Breakwell's Seedling, and the seven of Frederick. However, some nominal replicate samples did not have identical SNP profiles to each other (Fig 4 and S1 File: spreadsheet '*Samples with Nominal Reps*'). For instance, of the 125 varieties for which we had two identically named samples, SNP profiles were identical for 90 (72.0%) of the pairs (Table 3). In 11 cases (8.8%), nominal replicates had one SNP difference and in two cases (1.6%) the replicates had two SNP differences (in the bootstrapped dendrogram these pairs still clustered with bootstrap support of 100%). For 22 (17.6%) of the pairs, however, there were 4 or more SNP differences between them; indeed, for

**Table 3. The accessions with nominal replicates that had identical SNP profiles.**

| No. Reps | Same / Similar | | | Odd One | Different |
|---|---|---|---|---|---|
| | **Identical** | **1 SNP diff** | **2 SNP diffs** | | |
| 2 | 90 | 11 | 2 | - - - | 13 |
| 3 | 36 | 4 | 0 | 6 | 0 |
| 4 | 10 | 2 | 0 | 4 | 0 |
| 5 | 8 | 3 | 0 | 1 | 0 |
| 7 | 1 | 0 | 0 | - - - | 0 |
| 8 | 1 | 0 | 0 | - - - | 0 |
| 10 | 1 | 0 | 0 | - - - | 0 |
| 22 | 1 | 0 | 0 | - - - | 0 |

The first column is the number of times that named replicates had identical SNP profiles. The other columns indicate the number of times 'replicates' were not alike and by how many SNPs they differed. The 'Odd One' column indicates that one of the nominal 'replicates' had a different SNP profile to the other accessions with the same name. The last column indicates the number of nominal 'replicates' that differed by 4 or more SNPs; for these accessions we could not determine which, if either, was the 'true' genotype for the named variety. Of the 541 lines with replicates, 36 (6.7%) had SNP profiles that did not correspond with their names.

20 of these 22 pairs there were 8 or more differences (two pairs had 15 differences). Amongst these identically named pairs several came from the NFC. The NFC accessions had been genotyped previously [20, 21] and had been assigned unique MUNQ codes [11, 12] permitting a more objective assessment of which samples were true biological replicates; nine of the nominal replicates from the NFC were confirmed not to be so, the accessions in the pairs having distinct MUNQ codes (S1 File; spreadsheets '*Samples studied*' and '*Samples with Nominal Reps*'). Amongst the 47 varieties for which we had three nominal replicates, 36 (76.6%) had identical genotypes within the triples, in 4 cases (8.5%) one of the samples had one SNP difference from the other two (Table 3). In 6 cases, one of the triples had a very different genotype to the other two, and for the three samples named Strawberry Pippin, none of the samples were alike which was in concordance with the fact that each had a separate MUNQ codes. Differences were also recorded for varieties for which 4, 5 and 6 replicates were collected (Table 3, S1 File: spread sheet '*Samples with Nominal Reps*'). Before estimating the final error rate, MUNQ codes were used to remove genotypes known to be unique from the list of nominal replicates: 22 samples that were related as homonyms rather than being true, biological replicates were removed from the analysis leaving 541.

Using two contrasting assumptions about the fidelity of names assigned to the remaining samples, we estimated the overall SNP calling error rate (error rate was calculated as number of differences between identically named sample divided by the total number of SNP calls). In total, we had 563 accessions with nominal replicates, each having a maximum of 21 SNP calls; this makes a maximum of 11,823 SNP calls. However, there were 122 'no calls' (Fig 2) amongst these lines so that only 11,701 calls were made. Under the assumption that, having removed accessions known to be genotypically different, accession name could be used to predict replicate samples in the remainder—292 miscalls (errors) would have been made, which represents an error rate of (292/11,701) 2.5%. Under the alternative assumption that errors in sample names may still be present and that two identically named accessions with more than 2 SNP differences between them were not true replicates, we would need to remove 36 samples from the analysis (both samples of the 13 accessions with two replicates that didn't pair, and the odd one out for varieties with more than two replicates—10 in total) leaving 11,067 (11,823—(36 x 21)—112 = 10,955) calls; amongst these only 20 miscalls were made which represents an error rate of 0.18%.

## Ability of SNP panel to discriminate different varieties

If the names given to apple varieties were unique our SNP panel would have produced unique SNP profiles for all accessions collected as singletons. This was not the case, however; of the 1,919 accessions of nominal singletons, 424 (22.1%) had SNP profiles identical to at least one other accession (S1 File: spreadsheet '*Nominal Singletons with Reps*'); these 424 accessions formed 144 groups of indistinguishable SNP profiles (groups ranged from 2–23 accessions). Furthermore, the accessions in 16 of these groups had identical SNP profiles to accessions within known replicate groups; for example, 15 accessions had identical SNP profiles to the 10 replicates of Cox's Orange Pippin. At first sight, this might appear to indicate that our primer panel is not able to discriminate varieties. However, when the groupings were considered in the knowledge that a number of genotypes were replicated in the form of clones and sports, most of these issues could be resolved. Again, the MUNQ codes associated with the NFC accessions were helpful in this process: for example, the 15 accessions that were indistinguishable from Cox's Orange Pippin were confirmed to have already been allocated to the same MUNQ code (MUNQ 163); a group of 17 indistinguishable accessions, including Decosta, Jonagored and Wilmuta, had been allocated to MUNQ 901; and the accessions Bietighermer, Kirkes Lord Nelson and Tordai Alma, the names of which give no clue to their association,

had all been allocated to MUNQ 482 (S1 File: spreadsheet '*Nominal Singletons with Reps*'). Other clusters proved to be of reported synonyms (e.g., Fiesta and Red Pippin), colour mutants (Pixie and Red Pixie) or sports of a named variety (e.g., Barnack Beauty and Barnack Beauty sport). In a small number of cases, samples with very different MUNQ codes, for example Cartaut (MUNQ 754) and Charlot (MUNQ 342) and Roba (MUNQ 2538) and Sandringham (MUNQ 41), were found to have the same SNP profile. In the former case, it would seem likely that a sampling error has been made as the two accessions are in neighbouring plots of the Brogdale collection. The latter, on returning to the collection, proved to be a known error in repropagation that had not been reported to us; thus the two samples were of the same accession. In those cases where differently named accessions that had identical SNP profiles had been collected from small orchards or private gardens, and so hadn't been genotyped before and had no MUNQ code, we could not determine which of the named accession, if either, was the genuine one.

Interestingly, there were two situations in which the failure of a marker appears to be a genuine part of the SNP profile: eight accessions—Crowngold, Excel, Jonagored Supra, Jorayca, Josegold, Prince Jonagold, Red Jonaprince, Veekmans Jonaster and Wilmuta—thought to be mutations of Jonagold (all MUNQ 901) all share the absence of the marker BA08; the last four also have a missing BA19b; Holstein and the three mutation from it, Holstein Mahler, Holstein Palloks, Holstein sport (all MUNQ 803), all have BA06 missing.

## Heterozygosity and ploidy

It has been reported that there is a correlation between ploidy and heterozygosity; that is, triploids tend to have higher heterozygosity than diploids [13, 22, 23]. Amongst the 2,482 samples studied here, 294 (11.8%) were reported to be triploid and 31 (1.2%) reported to be tetraploids according to analysis with flow cytometry and or SSR markers [20]. Heterozygosity scores for the 2,482 samples ranged in value from 0.14 to 0.90, with an average of 0.47 (S1 File: spreadsheet '*Ploidy and Heterozygosity*'). Although there wasn't a precise division into triploids and diploids, the former tended to have high values for heterozygosity whilst the latter had low values. In fact, the highest value, 0.9, was for a triploid. Amongst the 200 accessions with the highest heterozygosity there were 101 (50.5%) triploids and 5 (2.5%) tetraploids. In the 200 accessions with the lowest heterozygosity, there were no (0%) triploids and 5 (2.5%) tetraploids. A one-way ANOVA ($F_{(3,2478)} = 206.3$, $p = 2.2\text{e-}16$) showed there to be a statistically significant difference between the mean scores for triploids (0.62) and both the diploids (0.44) and tetraploids (0.46). The unknown samples ranged in heterozygosity from very low to very high. On the dendrogram, triploids were more common in clusters 4 and 7 (20.1% and 17.8% of accessions within the respective clusters) than the other five clusters. Tetraploid samples did not cluster. The relationship between heterozygosity and ploidy is shown in Fig 5 and in S1 File: spreadsheet '*Ploidy and Heterozygosity*'.

## Discussion

In our attempt to develop a minimal marker set capable of discriminating all unique accessions held in U.K. collections, we considered that the markers chosen should fulfil various characteristics: they should be reliable with low fail and error rates; they should produce distinct genotypes for most, if not all, varieties; they must produce identical SNP profiles for replicates of the same variety. Finally, we would require that our SNPs be at least as good as microsatellite markers in distinguishing samples held in important, well curated collections. To determine the reliability of the markers, three different approaches were used. Firstly, we directly compared the genotype calls obtained with our 21 markers with those produced using the

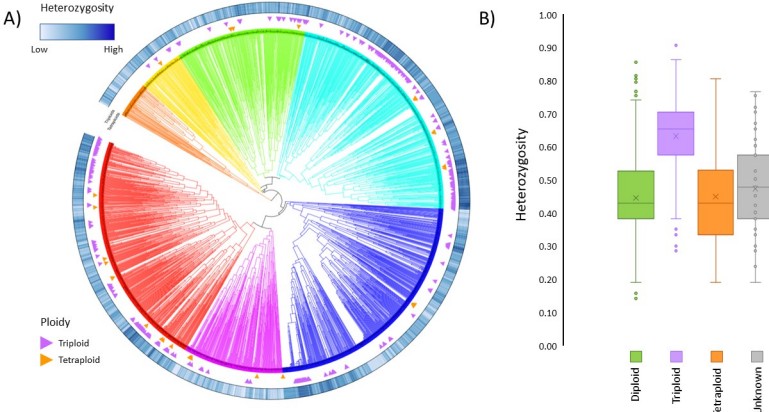

**Fig 5. A)** The relationship between heterozygosity and ploidy of apple accessions; the tracks around the dendrogram, from the outside to the inside, are i) heterozygosity, ii) triploids, iii) tetraploids. Most samples from triploids are to be found in clusters 4 (light blue) and 7 (red). **B)** Mean heterozygosity of samples grouped by ploidy; the triploids had a statistically significant different mean to that of the other three categories. The list of triploid and tetraploid accessions was taken either from the 2010 DEFRA report [20] or was reported by the curator of the orchard from which the accession was collected.

SeqSNP® platform. The disagreement between the scores given by the two platforms was very low. Overall concordance between the two platforms for all 21 markers was 95.8%; this is higher than the 94% concordance for a study of 96 tomato plants reported by those who developed the SeqSNP® platform [24]. For 19 of the 21 markers, however, concordance was more than 98%. Unfortunately, for the two markers that performed poorly, BA07b and BA22, we cannot determine whether the error in reporting the SNP was on the part of the SeqSNP® or KASP markers. However, it may be that these two markers can be replaced with two alternatives to obtain agreement across platforms. Second, after dimensionality reduction using clustering, a comparison was made between the highly supported clusters based on the SNP scores from the two platforms, SeqSNP® and KASP; these were essentially identical. Third, samples that were believed to be replicates (that is, they either shared the same name or, where available, the same MUNQ code) were tallied to determine how often they had identical genotypes; with a small degree of error, replicate samples had identical genotypes.

However, there were some errors evident; some of these may be the result of mislabelling of accessions or due to mishandling of samples but some will also be due to genotyping errors. There are many reasons why errors may occur in SNP scoring [25]. The most obvious source of error is related to the quality of the DNA; variation in quality could account for the failure of some of the markers. More specifically, KASP genotyping is based on the competitive hybridisation of two different fluorescently labelled probes to alternative alleles and the resolution of these based on both colour and intensity. The intensity ratios between samples and markers may differ slightly because of factors such as ploidy (Fig 2) or hybridisation efficiency [26]. For example, the relative intensity of the two fluorochromes, and thus the position of the sample on the plot, may well be different for a heterozygous diploid (genotype A:T) to that of a heterozygous triploid (genotype A:A:T or A:T:T). Also, additional polymorphisms surrounding the target SNP may reduce probe hybridisation efficiency resulting in decreased signal intensity from the fluorochrome and possibly to a null allele. Similarly, if the probe can hybridise to multiple regions within the genome, an increase in signal intensity may result. However, in

our experience, with good quality DNA, appropriately designed markers and well-trained SNP calling algorithms, error rates are low [27]. However, depending on the assumptions made, the error rate was between 0.18% and 2.5%. The latter is an overly conservative estimate and was based on the assumption, that proved to be false, that accession name could be used as a reliable indicator of genotype. It is well known that mislabelling can be found in plant collections; even in well curated collections, cultivar names are not always a reliable indicator of genotype, as was highlighted in the case of the cultivar name Topaz [11].

Interestingly, there were two situations in which the failure of a marker appears to be a genuine part of the SNP profile; that is, they may result from null alleles. Eight accessions—Crowngold, Excel, Jonagored Supra, Jorayca, Josegold, Prince Jonagold, Red Jonaprince, Veekmans Jonaster and Wilmuta—thought to be mutations of Jonagold (all MUNQ 901) all share the absence of the marker BA08; the last four also have a missing BA19b; Holstein and the three mutation from it, Holstein Mahler, Holstein Palloks, Holstein sport (all MUNQ 803), all have BA06 missing. In these cases, the null allele may be the result of variation (additional SNPs or deletions) in the primer binding sites for these markers.

Although we have identified potential errors, the overall low error rate gives weight to our claim that any samples of the same variety would have identical SNP profiles. Certainly, if technical replicates were to be included in any study, errors would be identifiable. Any large deviations between two samples believed to be from the same variety would almost certainly indicate errors in labelling or sample handling. If, as we maintain, error rates are low, any two samples that differ in their genotypes by more than one or two SNPs must be considered genuinely different. It would not seem unreasonable, therefore, to conclude that many samples that shared the same name were not, in fact, replicates of each other at all (S1 File: spreadsheet 'Samples with Nominal Reps'). Indeed, the use of MUNQ codes, where available, clearly demonstrated this.

Of the 125 named varieties for which two samples were apparently collected, 22 of the supposed replicates had 4 or more differences in their SNP profiles; many had more than 8 differences (e.g., Camelot, Le Bret and Ontario) and two had 15 (Melrose and Roundway Magnum Bonum). In such cases, it was not possible to determine which, if either, was the genuine representative of its type based on the SNP analysis alone. In some cases, there were obvious and identifiable errors in naming. For example, the two replicates of the variety EB54 had very different genotypes (8 differences) but the sample of EB52 was identical to one of the EB54 samples; this would appear to be a handling issue in our laboratory. Where we had SNP profiles for three or more samples of a variety the odd one out can be spotted easily. For example, the variety Gennet Moyle, for which, by name, we had four samples, three had identical genotypes whilst the fourth differed at 11 of the markers; this included two samples of Gennet Moyle from the NFC which had different MUNQ codes. In the case of samples named Strawberry Pippin, of which we had three, each had a distinct SNP profile which fully concurred with the fact that each of the accessions had different MUNQ codes.

Finally, we need to consider whether the varieties reported to have identical genotype profiles based on microsatellite data [12, 20] but that had different SNP profiles in our study are genuinely different from each other or represent mislabelled samples. As an example, we observed a small number of cases where a named variety was part of a large group in the DEFRA report [20] and, therefore, had the same MUNQ code as the others in the group, but was distinct in our study (e.g., Red Fameuse, Vegi Cox, Biesterfelder Renette, Calville Rouge du Mont d'Or, Fuji and Glengyle Red). The sample of variety Vegi Cox, that, using SSR markers, is indistinguishable from Cox's Orange Pippin [20], differed from it by 12 SNPs in our study; almost certainly, this is an example of a mislabelled sample. As a further example, Broad Eyed Pippin and Betty Geeson, which are reported to be indistinguishable with microsatellite

markers (both MUNQ 473), in our study differ from each other by one SNP call, marker BA14 reporting the former to be heterozygous and the latter to be homozygous; this may well be a SNP calling error as samples that lie between clusters, if not given a no-call, may be assigned to the incorrect cluster (Fig 3).

## Heterozygosity and ploidy

Given the small number of markers used in this study, it was somewhat surprising that we were able to detect a difference in heterozygosity between diploids and triploids. Indeed, this was not a major focus of our study as a clear signal would only be apparent with the use of a much greater number of markers [23]. However, It would be interesting to determine whether those diploid lines with high heterozygosity studied here will prove to be triploid. However, given that we have used a small number of SNP markers to determine this, there might be quite a marked sampling bias such that accessions that have high heterozygosity may not be triploid whilst accessions that have low heterozygosity may, indeed, prove to be triploid. The correlation between the two factors will also be weakened by mislabelled samples such that a diploid variety is incorrectly given the name of a triploid or *vice versa*. For example, of the four samples of Crimson King, a known triploid, only three had identical SNP profiles; these three had heterozygosity scores of 0.70. The sample that didn't cluster had a score for heterozygosity of only 0.29; it was almost certainly not Crimson King and not a triploid. Nonetheless, polyploidy can often be identified with reasonable certainty from SSR markers through the regular presence of third and fourth alleles (although confirmation by cytology is still valuable). Larsen et al. [22] reported a clear distinction in the level of heterozygosity of triploid accessions using data from over 15,000 genome-wide SNPs. The measure of heterozygosity here would not be sufficient to provide confidence in attributing ploidy, but it would offer a useful screening criterion in order to select individuals for cytological, or other analysis. We report that the triploid samples in our analysis were more commonly placed in clusters 4 and 7. However, a study based on DArT markers didn't report specific clustering by ploidy type [21]. Heterozygosity of tetraploids was not different to that of diploids. This probably is a reflection of the fact that most tetraploids are thought to be autopolyploid arising from doubling of the diploid genome rather than combining of two different genomes as is thought to be the case for triploids.

## Development of marker systems for genetic analysis

Although the set of microsatellites currently used for apple identification are highly polymorphic and able to distinguish between the majority of, although not all, apple varieties [20], they have a number of disadvantages for a contemporary marker system [28]; for instance microsatellite analysis is difficult to automate, especially with regard to data capture and scoring. Indeed, many breeders and scientists working on agronomically important crops have moved away from using microsatellites and are now using SNPs. We believe that, by using SNP markers, we have produced a dataset with a degree of future proofing. SNP-based genotyping is inexpensive (£5 per sample) compared to SNP-array-based genotyping and results can be easily record results in a spreadsheet. For KASP genotyping the most cost-effective experimental design employs a large number of samples with a relatively small marker set. For example 20–24 markers may be screened against 96 samples for £300 (£3.13 per sample), however, by increasing the number of samples to 384 reduces the cost per sample to £1.37. Hence, a re-cataloguing of the whole apple collection might prove to be quite inexpensive. Finally, may well be that for such a SNP panel 21 markers is a little too few. A slightly higher number of markers, perhaps 25 rather than 21, would probably be more robust and allow for a small number of failed markers without running the risk of confusing two similar but none-identical accessions.

## Supporting information

**S1 File. Excel workbook containing six spreadsheets.** 1. Samples Studies, 2. Nominal Single-tons; 3. Nominal Singletons with Reps; 4. Samples with Nominal Reps; 5. Ploidy and Heterozygosity; 6. KASP probes.
(XLSX)

**S1 Fig. Dendrogram of all samples studied.**
(PDF)

## Acknowledgments

We are grateful for contribution of samples from the following orchards: National Fruit Collection, Brogdale, Botanic Gardens of Wales, and various small orchards in the Bristol area: Field Farm Nursery, Frank Matthews Orchard, Gogerddan Welsh Heritage Museum Orchard, Goldney Hall Orchard, Horfield Organic Community Orchard, Linden Lea Orchard, Metford Road Community Orchard, Shepton Mallet Cider Apple Collection, St. Monica's Orchard, Thatchers' Exhibition Orchard, and Worle Nursery. We would particularly like to thank Caroline Denancé, Hélène Muranty and Charles-Eric Durel (IRHS, INRAE, France) for the supply of MUNQ codes across the whole NFC collection.

## Author Contributions

**Conceptualization:** Mark Winfield, Keith Edwards, Gary Barker.

**Data curation:** Mark Winfield, Matthew Ordidge, Danny Thorogood.

**Funding acquisition:** Helen Harper, Keith Edwards.

**Methodology:** Amanda Burridge, Paul Wilkinson.

**Software:** Mark Winfield.

**Writing – original draft:** Mark Winfield.

**Writing – review & editing:** Mark Winfield, Amanda Burridge, Matthew Ordidge, Helen Harper, Paul Wilkinson, Danny Thorogood, Liz Copas, Keith Edwards, Gary Barker.

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
