## [Decision Letter · Decision Letter 0]

9 Sep 2020

PONE-D-20-23044

Development of a minimal KASP marker panel for characterising apple collections

PLOS ONE

Dear Dr. Winfield%,

Thank you for submitting your manuscript to PLOS ONE. After careful consideration, we feel that it has merit but does not fully meet PLOS ONE’s publication criteria as it currently stands. Therefore, we invite you to submit a revised version of the manuscript that addresses the points raised during the review process.

We look forward to receiving your revised manuscript.

Kind regards,

Tzen-Yuh Chiang

Academic Editor

PLOS ONE

Journal Requirements:

2. In your Methods section, please provide additional information regarding the field permits you obtained for the work. Please ensure you have included the full name of the authority that approved the field site access and, if no permits were required, a brief statement explaining why.

3. Please upload a new copy of Figure 3 as the detail is not clear. Please follow the link for more information: https://blogs.plos.org/plos/2019/06/looking-good-tips-for-creating-your-plos-figures-graphics/

Reviewers' comments:

Reviewer's Responses to Questions

**Comments to the Author**

1. Is the manuscript technically sound, and do the data support the conclusions?

Reviewer #1: Partly

Reviewer #2: Partly

2. Has the statistical analysis been performed appropriately and rigorously? 

Reviewer #1: I Don't Know

Reviewer #2: Yes

3. Have the authors made all data underlying the findings in their manuscript fully available?

Reviewer #1: Yes

Reviewer #2: Yes

4. Is the manuscript presented in an intelligible fashion and written in standard English?

Reviewer #1: Yes

Reviewer #2: Yes

5. Review Comments to the Author

Reviewer #1: Overall it is interesting work and it was a pleasure to review. In my opinion, a major review is necessary to consider the publication.

Comments

LINE 104 -(Harper et al., 2019)., reference number?

LINE 293: Fernandez-Fernandez, 2010, reference number?

Line 296: Fernandez-Fernandez, 2010, reference number?

LINE 323: Larsen et al (2018), reference number?

LINE 354: 2,675 trees representing more than 2,200 distinct accesions: 2,104 (National Fruit Collection at Brogdale in Kent) + 153 (National Botanic Garden of Wales) + 504 various orchards and gardens = 2104+153+504 = 2,761 trees?.

LINE 42: 2,143 lines from the National Fruit Collection (NFC) , 2,143 or 2,104?

The number of accessions studied is a bit confusing throughout the text

LINE 352 Experimental Procedures: Why is the Experimental Procedures section not put as material and methods before the results? the understanding of the article as a whole would be clearer

LINE 360: “For 229 varieties, samples were taken from more than one listed under the variety name (often these 362 were from different orchards or gardens)”. In Sup. File 1 I see: 125*2 + 47*3 + 16*4 + 12*5 + 1*7 + 1*8 + 1*10 + 1*23 = 204 varieties. I suppose that the rest of the varieties have failed, but it is not clear, a basic table is necessary where the collected material is seen very clearly, the number of accessions of each collected variety, and the material that has failed, with final sums where You can quickly and clearly see the different numbers of samples that are discussed throughout the paper. To be understandable, everything must be detailed in a single table, because otherwise it makes understanding very difficult

LINE 377: “markers capable of differentiating all 380 cultivars used in that study”. The 380 cultivars are those studied by Harper (2019)? because line 359 talks about 372.

LINE 387-388: “For each of the 21 markers, two allele-specific forward primers and one common reverse 388 primer were designed using the same primer sequence used in Harper et al. (2019). (S1 File: 389 ‘KASP probes as used’).”. looking at the table, it is seen that there are 31, I understand that the 10 that do not have a name are those that Harper used in his work, and that they are not used in the present work, but it is necessary to clarify it in some way, it does not make sense to see some markers, without a name or anything, you don't know what it is

LINE 404:Estimating error rate and identification of mislabelled samples:

very important aspect . It is very difficult to understand the permissible error. It is an aspect that I do not understand at all, especially in species such as the apple tree, where the original mutations are very frequent, and that has given rise to numerous varieties that present very few, but visible, differences with the originals.

When studies with SSRS, a difference between two accessions in a single microsatellite is already enough to consider them as different accessions or varieties

I understand that the samples with repetitions or at least some of them, have also been studied with microsatellites (line 417: “Most of the accessions obtained from

Brogdale and the Welsh Collection have been genotyped previously using microsatellite or DArT markers or both”). Please show the comparison between microsatellites and SNPs. Those samples that show variation in Snps, did they show an identical profile using microsatellites or did they already show some variation?

This is a very important and fundamental aspect, especially when identifying close varieties, due to the high mutation rates present in apple trees, with SSRS a difference in a single marker already leads to consider that they are two different varieties.

Of the 204 accessions that present repeats, how many have coincided for the total of 21 SNPs?

Please explain this aspect very clearly, and compare the data with the results obtained through SSRs.

See for example Analysis of the genetic diversity and structure of the Spanish apple genetic resources suggests the existence of an Iberian genepool (Pereira et al, 2017)

https://onlinelibrary.wiley.com/doi/abs/10.1111/aab.12385 : in page 431 I see the following: “J1 was a main group of cultivars related with ‘White Astrakhan/Papirovca’ (checked at EMR by Felicidad Fernández, personal communication) at JC=0.18, in which ‘Transparente Blanca’ (only differed by one allele to ‘White Astrakhan’) and ‘Mutsu’ were clustered; all of them belonged at RPP2 with a qI >80%.” both varieties are distinguished by a single allele, so I am unable to understand how two accessions that show any difference can be taken as synonyms by Snps

Or see Genetic diversity and structure of local apple cultivars from Northeastern Spain assessed by microsatellite markers (Urrestarazu et al,2012) https://rd.springer.com/article/10.1007%2Fs11295-012-0502-y “Within the 304 unique genotypes, small profile differences involving a single allele were found among 16 accessions (Table 5). These minor differences can be due to spontaneous somatic mutations that are known to occur in long-lived trees propagated by grafting (Weissinger 1985).”

There are 8 groups of accessions that differ by a single allele, considering mutations

Please, clearly develop this point

LINE 83: “One can adjust for these experimental errors when samples have known replicates, e.g. the sample Black Vallis 1 differed by one SNP (BA14) from the other two Black Vallis samples creating two distinct genotypes where we presume there should be only one”

the same problem as before, why is it known that they are the same? showed the same profile by SSRs? If so, how reliable are the SNPs considering two accessions that differ by a marker, when they are equal by SSRs, and also knowing that there are varieties that differ only by one allele, and that the species mtation rate is elevated? Is it a system that can be used with total reliability with that margin of error that is not found with SSRs? Is hard to understand

LINES 113-123 “we expected complete concordance between the SNP profiles produced by the two assays”…..” the overall concordance between the two platforms for all 21 markers was 95.8%”

I am understanding from this explanation that the technique is reproducible in a percentage of 95.8. Is correct? How would the case of a litigation on a plant patent with a very small genetic difference be resolved if the reproducibility is not 100%?

LINES 135- 168: Accuracy of scoring replicates.

It is not well understood, the replicate samples, had they been tested by SSRs, and by SSRs they were the same and now they do not match? and those that do not match are removed from the analysis? or using SSRs they did not match and there were differences between them?

LINE 145-150. How is it possible that two identical samples do not show the same profile and what reliability can be given to the analysis when entering a new sample to identify it? because there will be more unrelated samples that differ only by two Snps, and are not considered equal, when will we consider them equal if we do not have the names?

Stated in another way, if it was not known that they were replicated a priori, they would be considered synonymous samples showing two different Snps if the previous information was not had??

LINES 193-196: “At first sight, this might appear to indicate that our primer panel is not able to discriminate varieties. However, when the groupings were considered in the knowledge that a number of genotypes were replicated in the form of clones and sports, most of these issues could be resolved.”

What does the method contribute if one does not start from previous information on the accessions? If we start from 1000 accessions for which we have no previous information, how sure are we that we make a correct identification and that all synonyms or mutations are identified?

LINES 206-208: “In those cases where differently named accessions that had identical SNP profiles had been collected entirely from small orchards or private gardens, no conclusions about the groupings could be drawn.”

Are they not considered synonymous if they present the same SNPs or by microsatellites were they different? or the information with 21 SNPs is not enough to establish that they are true synonyms?

LINE 210: Heterozygosity and ploidy Why is ploidy not known out of 92 samples?

LINE 242-245: “Third, samples that were believed to be replicates (that is, they either shared the same name or, where available, the same MUNQ code) were tallied to determine how often they had identical genotypes; with a small degree of error, replicate samples had identical genotypes”

Is this error in the identification of collections permissible, and especially in the event that a legal conflict arises about different similar varieties, and that they only differ due to a somatic mutation, or is this method not recommended for that purpose?

LINES 270-272: “If, as we maintain, error rates are low, any two samples that differ in their genotypes by more than one or two SNPs must be considered genuinely different”

With SSRs with a difference in a micro, two varieties are already considered, why not with SNPs?. If we have two unknown samples, or named differently, and they differ by a single SNP, are they also considered genetically identical even though they were not collected with the same denomination?

S1 Fig. Dendrogram of the 2,482 apple accessions studied. I can't find that figure. however, a Supplementary_File_1.pdf file appears. Could it be an error loading the files?

The tables should be clearer and more concise, what do the colors that appear in them mean?, for example, in table “Accessions Studied” What relevant information do the ”Notes (Matt Ordidge)” and “Notes (Mark Winfield)” columns provide? What does it mean that some samples are colored orange and others yellow?

In the table “Ploidy and Heterozygosity” can't remove columns at the end with irrelevant data or no data?

2- Do the authors think with certainty that from the merely agronomic and identification point of view, without taking into account the economic cost, SNPs provide more solid and robust information than SSRs?

Can it be categorically stated that only using 21 SNPs allows correct identification of all varieties be made, detecting somatic mutations? Is it possible to correctly identify and classify the varieties without having any prior information about them?

How would the case of a dispute over a plant patent with a very small genetic difference be resolved? would it be possible with 21 SNPs?

Reviewer #2: This manuscript describes interesting results about the selection and validation of a set of 21 SNP markers for distinguishing genotypes in a large UK apple collection using the KASP technology. The article is well written and sufficiently detailed. The results seem robust. The discussion is sound even if some additional points could be further commented on. In the Experimental procedures, for each BAxxx KASP marker, the corresponding SNP_ID as defined by Bianco et al. (2016) for the Axiom®_Apple480K array from which the SNP have been selected through Harper et al’s work (2019) should be detailed in order to allow complete tracing of the correspondence by the readers.

Additional comments and advices are detailed in the attachment.

6. PLOS authors have the option to publish the peer review history of their article (what does this mean?). If published, this will include your full peer review and any attached files.

Reviewer #1: No

Reviewer #2: No

---

## [Author Response · Author response to Decision Letter 0]

28 Sep 2020

We have included a letter, 'Response_to_Reviewers.docx', as requested.

---

## [Decision Letter · Decision Letter 1]

23 Oct 2020

PONE-D-20-23044R1

Development of a minimal KASP marker panel for distinguishing genotypes in apple collections

PLOS ONE

Dear Dr. Winfield,

Thank you for submitting your manuscript to PLOS ONE. After careful consideration, we feel that it has merit but does not fully meet PLOS ONE’s publication criteria as it currently stands. Therefore, we invite you to submit a revised version of the manuscript that addresses the points raised during the review process.

We look forward to receiving your revised manuscript.

Kind regards,

Tzen-Yuh Chiang

Academic Editor

PLOS ONE

Reviewers' comments:

Reviewer's Responses to Questions

**Comments to the Author**

1. If the authors have adequately addressed your comments raised in a previous round of review and you feel that this manuscript is now acceptable for publication, you may indicate that here to bypass the “Comments to the Author” section, enter your conflict of interest statement in the “Confidential to Editor” section, and submit your "Accept" recommendation.

Reviewer #1: (No Response)

Reviewer #2: (No Response)

2. Is the manuscript technically sound, and do the data support the conclusions?

Reviewer #1: Yes

Reviewer #2: Yes

3. Has the statistical analysis been performed appropriately and rigorously? 

Reviewer #1: Yes

Reviewer #2: Yes

4. Have the authors made all data underlying the findings in their manuscript fully available?

Reviewer #1: Yes

Reviewer #2: Yes

5. Is the manuscript presented in an intelligible fashion and written in standard English?

Reviewer #1: Yes

Reviewer #2: Yes

6. Review Comments to the Author

Reviewer #1: Congratulations on the great proofreading job. I have understood the approach very well. I still have some points that are not very clear and some little corrections

• Thank you very much for all the clarifications, I am still not very clear about the identification issue, please, if you can clarify it for me a little more and on a personal level I appreciate it: if I provide you with 100 samples of which I do not know their origin, not your name or anything to identify them to me, do you think that SSRs, SNPs or both are more reliable? To consider two synonymous samples, would it be necessary for the 21 SNPs to coincide or would an error be allowed? Or is it that in the case of a difference in a single SNP the analysis would be repeated to make sure? Thanks

• A point that I do not understand well, if you can clarify it for me. In the File_S1 sheet "Accessions with Nominal Reps", I see that there are samples with the same MUNQ code, but different for SNPs, those samples were the same for SSRs and are they different for SNPs? (for example Afal and Cwmafan 1 and 2). This is considered an experimental error and would it be repeated? This is where the question arises again, if we did not know that they had the same name, would the analysis also be repeated? is the point that I can't quite understand

I think that from the first moment it should be made clear how many samples have been eliminated for the analysis, and from there always talk about the samples included in the analysis, because otherwise in some places it is confusing

LINE 41: “To test the efficiency of these markers we have used them to characterise circa 2,200 distinct apple varieties: 2,104 accessions from the National Fruit Collection (NFC) at Brogdale, Kent, and 143 from the Welsh Botanical Gardens at Carmarthen”: The accessions are still not very clear here. Accessions and varieties are mixed. 2200 varietes, 2104 accessions, 143 accessions? Please, see if you can correct so that all are accessions or varieties, and that the sum matches

LINE 58: 2675 TREES; (Brogdale 2,104+ Llanarthney 143+VARIOUS 429 =2676). I still do not understand the number of samples; in Table S1 2482 accessions appear collected. What do the 193 missing accessions correspond to?

LINE 64: “For 2,058 of these, a single tree was sampled. For 229 varieties, samples were taken from more than one listed under the variety name (often these were from different orchards or gardens)”. Are these 2058 indicated in any excel sheet?. 229 are the repetitions collected in the sheet "Accessions with Nominal Reps" (because (2rep* 125) + (3 rep * 47) + (4 rep* 16) + (5 rep*12) + (7 rep *1) + (8 rep *1) + (10 rep *1) + (23 rep *1) is not 229 or am I misunderstanding references to samples and repeats

LINE 66: “From the cultivar Yeovil Sour, 24 replicates were taken from a single tree to test reproducibility”. In the table 23 appears, which is correct?

I understand that these numbers are after eliminating the samples that gave error, if so, it is interesting to clarify it here because otherwise it leads to confusion

Reviewer #2: The manuscript has now been significantly upgraded.

I consider it can be accepted after adressing some minor corrections/modifications.

7. PLOS authors have the option to publish the peer review history of their article (what does this mean?). If published, this will include your full peer review and any attached files.

Reviewer #1: No

Reviewer #2: No

---

## [Author Response · Author response to Decision Letter 1]

29 Oct 2020

We have included a rebuttal letter that includes a detailed response to the specific comments of the two reviewers.

---

## [Decision Letter · Decision Letter 2]

12 Nov 2020

Development of a minimal KASP marker panel for distinguishing genotypes in apple collections

PONE-D-20-23044R2

Dear Dr. Winfield,

We’re pleased to inform you that your manuscript has been judged scientifically suitable for publication and will be formally accepted for publication once it meets all outstanding technical requirements.

Kind regards,

Tzen-Yuh Chiang

Academic Editor

PLOS ONE

Additional Editor Comments (optional):

Reviewers' comments:

Reviewer's Responses to Questions

**Comments to the Author**

1. If the authors have adequately addressed your comments raised in a previous round of review and you feel that this manuscript is now acceptable for publication, you may indicate that here to bypass the “Comments to the Author” section, enter your conflict of interest statement in the “Confidential to Editor” section, and submit your "Accept" recommendation.

Reviewer #1: All comments have been addressed

Reviewer #2: All comments have been addressed

2. Is the manuscript technically sound, and do the data support the conclusions?

Reviewer #1: Yes

Reviewer #2: Yes

3. Has the statistical analysis been performed appropriately and rigorously? 

Reviewer #1: Yes

Reviewer #2: Yes

4. Have the authors made all data underlying the findings in their manuscript fully available?

Reviewer #1: Yes

Reviewer #2: Yes

5. Is the manuscript presented in an intelligible fashion and written in standard English?

Reviewer #1: Yes

Reviewer #2: Yes

6. Review Comments to the Author

Reviewer #1: (No Response)

Reviewer #2: This new version is very good now.

Thank you for taking into account all the comments raised in my last review.

Just a minor (but important) point: it seems that there is a typo in the sheet ‘KASP probes’ of supplementary Fig S1: the Axiom ID of the last KASP marker (BA31) is "AX-115515922" and not "AX-11551592" (the last ‘2’ is missing). Please check. Best.

7. PLOS authors have the option to publish the peer review history of their article (what does this mean?). If published, this will include your full peer review and any attached files.

Reviewer #1: No

Reviewer #2: No

---

## [Editor Report · Acceptance letter]

16 Nov 2020

PONE-D-20-23044R2 

Development of a minimal KASP marker panel for distinguishing genotypes in apple collections 

Dear Dr. Winfield:

I'm pleased to inform you that your manuscript has been deemed suitable for publication in PLOS ONE. Congratulations! Your manuscript is now with our production department. 

Kind regards, 

on behalf of

Dr. Tzen-Yuh Chiang 

Academic Editor

PLOS ONE